# LLMs as Research Tools: A Large Scale Survey of Researchers' Usage and Perceptions

**Zhehui Liao[1], Maria Antoniak[2], Inyoung Cheong[3], Evie Yu-Yen Cheng[4], Ai-Heng Lee[4], Kyle Lo[4], Joseph Chee Chang[4], Amy X Zhang[1]**

[1]University of Washington, Paul G. Allen School of Computer Science & Engineering
[2]University of Colorado Boulder, Department of Computer Science
[3]Princeton University, Center for Information Technology Policy
[4]Allen Institute for Artificial Intelligence

## Abstract

The rise of large language models (LLMs) has led many researchers to consider their usage for scientific work. Some have found benefits using LLMs to augment or automate aspects of their research pipeline, while others have urged caution due to risks and ethical concerns. Yet little work has sought to quantify and characterize how researchers actually use LLMs and why or why not. We present the first large-scale survey of 816 verified research article authors to understand how the research community leverages and perceives LLMs as research tools. We examine participants' self-reported LLM usage, finding that 81% of researchers have already incorporated LLMs into aspects of their research workflow. We also find that some traditionally disadvantaged groups in academia (non-white, junior, and non-native English speaking researchers) report higher LLM usage and perceived benefits, suggesting potential for improved research equity. However, women, non-binary, and senior researchers have greater ethical concerns. Our study provides much-needed evidence, rather than speculation, about how LLMs are currently being used as research tools.

## 1 Introduction

The burst in popularity of widely available generative AI tools, and findings from recent small-scale studies with researchers (Morris, 2023; Fecher et al., 2023) suggest that many in the research community have already found benefits in incorporating large-language models (LLMs) into their research workflows. Adopting these tools has opened up many possibilities, such as improved efficiency, greater research equity, and inspiring novel ideas. At the same time, new tools raise familiar research risks and ethical concerns— like transparency, reproducibility, plagiarism, and data fabrication—while introducing new dangers to the research process. Differences in perceptions about risks, ethics, and social acceptability across demographic groups and researcher backgrounds could also drive differences in adoption, so that any benefits accrue unevenly and exacerbate existing structural barriers in academia (Goyes and Skilbrei, 2023).

Due to high interest in the research community in developing and adopting LLMs as research support tools, there have been a plethora of recent work on investigating researchers' usage of LLMs. However, most has focused on one specific research domain; for example, HCI (Kapania et al., 2024), psychology (Ke et al., 2024), machine learning (Russo et al., 2024), and management research (Williams et al., 2023). While there have also been a few investigations that included multiple disciplines, they relied on small-scale interviews (Morris, 2023, N=20), surveys (Fecher et al., 2023, N=52), or automated textual analysis of published papers (e.g., Kobak et al., 2024; Liang et al., 2024b).

In contrast, our work represents the first large-scale survey (N=816) [1] focusing on researchers' use of LLMs across disciplines, providing empirical evidence for trends previously only speculated about. By asking researchers about both their general perceptions and personal usage, our survey allows us to describe developing norms in more detail, across disciplines and demographic groups. Participant recruitment can be challenging for large-scale surveys but crucial for the validity of the results. For this, our participants were sourced from Semantic Scholar, which maintains a verified repository of more than 100 thousands published researchers from a wide range of research domains, demographics, and research experiences. Our survey focused on understanding how researchers are *actually* using LLM-based researcher tools in their work today, and how they perceive the risks and benefits of leveraging LLMs for different research tasks. In particular, we were interested in researchers' perceptions about the acceptability of these tools and possible demographic differences, leading us to recruit researchers across nationalities, languages, career stages, disciplines, genders, and ages. The differences we uncover between these groups reveal rapidly changing social norms around the usage of AI tools in research, highlighting important considerations around research equity and broader adoption. Our contributions include the following.

- We find around 81% of respondents have used LLMs for research, with the tasks of information seeking and editing reported most frequently and data analysis and generation reported least frequently.
- Researchers who are non-white, non-native English speaking, and junior researchers both use LLMs more frequently and perceive higher benefits and lower risks, but women and non-binary researchers have greater ethical concerns.
- While LLMs are used across all fields, computer science researchers showed greater comfort with disclosure of usage and lower ethical concerns than other disciplines.
- Researchers generally prefer to use LLMs from open source/non-profit entities.

Our findings suggest that LLMs might help improve research equity for people from demographic groups traditionally facing certain structural barriers, according to the researchers from those groups. However, ethical issues around LLMs need to be confronted and social norms established within each field for broader adoption.

## 2 Related Work

**LLMs as Research Support Tools**  The AI and HCI research communities have been exploring research tools powered by LLMs to support all stages of the research workflow, from research ideation (Gero et al., 2022; Liu et al., 2024; Wang et al., 2024), paper reading (Lo et al., 2023; Fok et al., 2023), literature review (Lee et al., 2024b; Kang et al., 2023; Hsu et al., 2024), writing (Kim et al., 2023; Long et al., 2023; Gruda, 2024), peer review support (Sun et al., 2024; D'Arcy et al., 2024), and more. Additionally, there has also been increased commercial interests in building LLM-based tools for science, such as Galactica (Taylor et al., 2022), Perplexity, Elicit, OpenAI and Gemini's Deep Research modes, and many more. There are even attempts to build fully autonomous end-to-end research agents based on LLMs such as Sakana AI (Castelvecchi, 2024a;b) and FutureHouse (Sam Rodriques, 2024). These recent works show promise for LLM-powered research support tools, and our survey further confirms already widespread adoption in the research community.

**Risks and Ethical Implications of LLMs in Research**  There are many risks involved in using LLMs for research, including insufficient precision and accuracy (Alvarez et al., 2024), with a phenomenon of "hallucination" (Alkaissi and McFarlane, 2023)—generating plausible-sounding but fictional content. For example, Galactica, an LLM trained on scientific papers (Taylor et al., 2022), was taken down after producing convincing but false scientific articles (Heaven, 2022). Without vetting, inaccurate content could contribute to

---

[1]Link to survey questions, analysis code, and results: https://github.com/allenai/llm-research-survey.

misinformation and erode trust in research (Kobiella et al., 2024; Gruda, 2024; Antoniak et al., 2024; Toma et al., 2023). These issues are particularly concerning because many researchers, particularly those lacking expertise or resources, predominantly rely on commercial closed models (Wulff et al., 2024; Toma et al., 2023).

**Demographic Influences on LLM Perception and Adoption**    Individual perceptions and usage patterns of LLMs are shaped by various factors, including personality traits, age, gender, and educational background (Jakesch et al., 2022). For instance, people with a high level of agreeability and younger people tend to have more positive views of AI, while those susceptible to conspiracy theories often have more negative perceptions (Stein et al., 2024). Also, a notable gender gap has been observed in LLM adoption, with men users outnumbering women users (Draxler et al., 2023). LLMs might reduce certain inequities in research and publishing by lowering barriers for non-native English speakers (Morris, 2023) and providing high-quality reviews to novice researchers (Chamoun et al., 2024).

## 3 Methods

### 3.1 Survey Design, Participant Recruitment, and Data Collection

Through literature reviews and a formative survey on X/Twitter, we iteratively designed a questionnaire with feedback from researchers (see Appendix A.1). The survey asks about perceptions and usage of LLMs for different research tasks, with responses provided via a Likert rating[2] and a free response rationale. We recruited verified authors with at least one published paper listed on Semantic Scholar that has a dedicated data quality team maintaining academic author profiles. The study was reviewed and exempted by the IRB of the University of Washington. A recruitment email was sent to 100,187 verified authors, and 34,922 (34.9%) emails were opened. From November 2023 to April 2024, we collected 1,226 unfiltered survey responses via Qualtrics, which we filtered to exclude participants who did not progress past the first page or spent fewer than 2 seconds on each question. **This resulted in $n = 816$ total responses**.

We collected fine-grained self-reported demographic information and then manually coded and categorized the responses (see Appendix A.3). Of the 816 total responses, 644 included demographic information, with the following distributions. **Gender**: Man (79%); Woman, Non-Binary, Other (21%); **Race**: White (61%); Non-White (39%); **Years of Research Experience**: 11+ (57%); 4-10 (32%); 0-3 (11%); **Native Language**: Native English (62%); Non-Native English (38%); **Field of Study**: Computer Science (40%); Social Science & Humanities (24%); Natural Science & Engineering (21%); Biology & Medicine (15%). We provide additional details in Table 2 in the Appendix.

### 3.2 Quantitative Analysis

Each participant is labeled with (up to) five demographic categories and contributes (up to) 36 Likert ratings (an LLM Usage Frequency question and five LLM Perception questions, each repeated for six LLM Usage Types). To address *repeated measures*, we fit linear mixed-effects models (lme4 in R) of the form

$$\text{LikertRating}_{ij} = \beta_0 + \gamma_i + \beta_1\text{Demographic}_i + \beta_2\text{UsageType}_j + \epsilon_{ij} \tag{1}$$

to test the association between participant $i$'s LikertRatings (answers to LLM usage or perception survey questions) and participant Demographic fixed-effects, while controlling for participant-specific random effects $\gamma_i$ and the type of LLM usage in question $j$.

In total, we fit 30 models (6 types of Ratings x 5 Demographic groups), use likelihood ratio tests for significant Demographic effect ($H_0 : \beta_1 = 0$), and use Holm-Bonferroni (p.adjust in

---

[2]LLM usage is rated on a 6-pt scale while perception is rated on a 5-pt scale excluding an option "I have never used LLM for this type of activity" for later analysis.

R) to correct for multiple comparisons within model fits for the same Rating type. Each model was fit on approximately three to four thousand participant ratings after filtering out missing data; exact sample sizes per regression are in Table 4 in Appendix B. Given a significant result, we then also conduct post hoc tests for significant pairwise differences (emmeans() in R) in mean Rating between Demographics; for example, testing $H_0 : \mu_{\text{white}} - \mu_{\text{non-white}} = 0$. All survey questions, data, and R code is available in Supplementary Materials.

### 3.3 Qualitative Analysis of Free-Text Responses

To collect deeper insights, we analyzed free-text responses through an iterative open thematic analysis approach (Boyatzis, 1998; Connelly, 2013). Three of the paper authors read through and annotated the same subset of the responses into thematic categories independently. Through discussion with the entire research team and additional rounds of independent annotation, we settled on a final set of themes. The same three authors annotated the full set of responses (each response was annotated by one author). The full set of themes (one set per question), with definitions and examples, can be found in Appendix C, and we release our full set of annotations along with our other data and resources.

## 4 Results

### 4.1 How do researchers use LLMs?

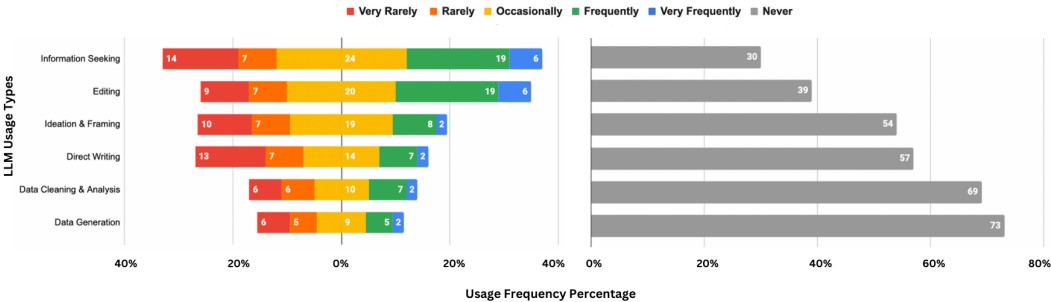

Figure 1: The left diverging bar chart displays the distribution of usage frequency across six types of LLM usage. The midpoint (0%) is centered at "Occasionally." The grey bar chart indicates percentages of responses that report "Never" using LLMs for each type.

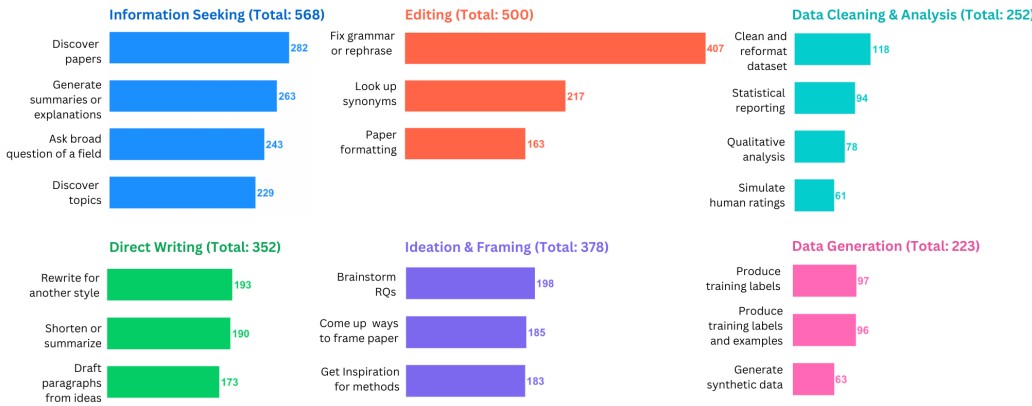

Figure 2: Each bar chart shows the number of participants who reported using LLMs for the indicated research task. Participants could select multiple tasks and subtasks.

We asked participants to mark how frequently they used LLMs for six types of usage: Information Seeking, Editing, Ideation & Framing, Direct Writing, Data Cleaning & Analysis, and Data Generation. Overall, we find that LLMs are frequently used by researchers, with **80.88% (660 out of 816) of respondents reporting some use**. The most popular tasks were Information Seeking and Editing (49% and 45% respondents rated at least occasional usage), with by far most usage on rewriting text to fix grammar or awkward phrasings (Figure 1 Figure 2). Most respondents (69% and 73%) stated they never used LLMs for Data Cleaning & Analysis or Data Generation.

We find that **researchers' racial identity is significantly associated with LLM usage**, with non-white researchers reporting more frequent usage of LLMs than white researchers ($p < .0001$).[3] When using LLMs for editing, we see significantly greater usage by non-native English-speaking researchers ($p < .0001$), though this difference was not found for other tasks (see first column in Figure 3).

## 4.2 How do researchers perceive LLM usage?

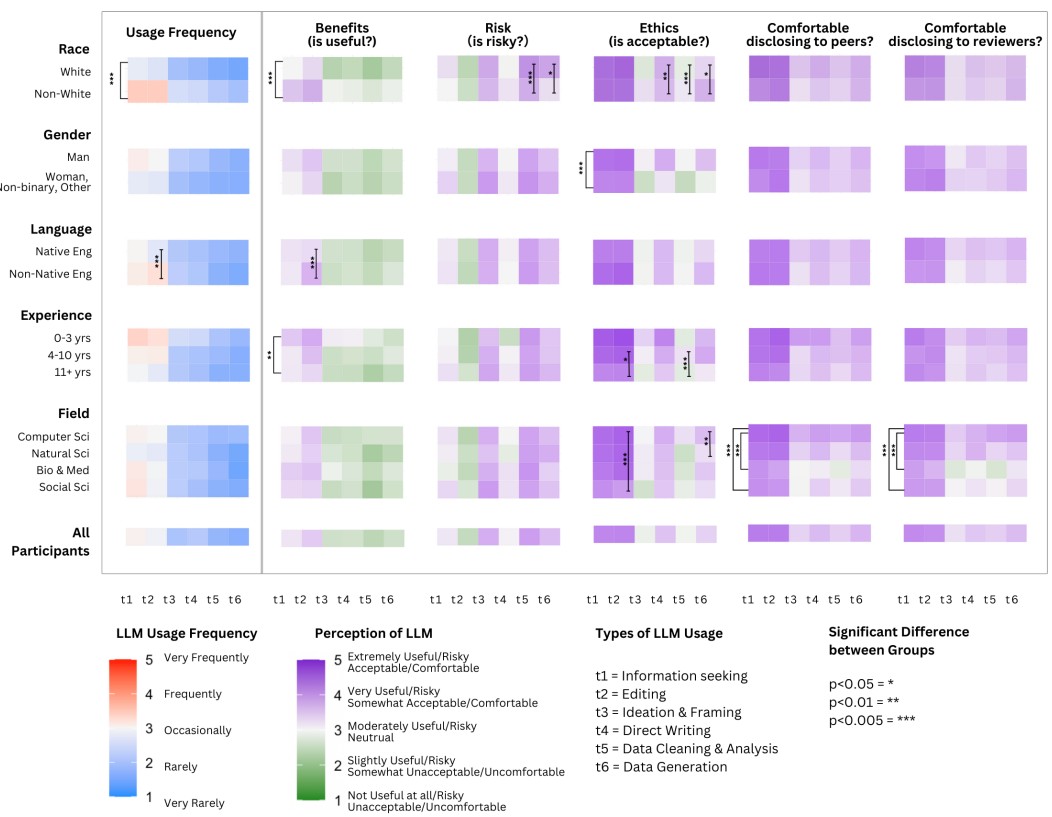

Figure 3: Each heat map square represents the **average** rating of this demographic group on the usage frequency or perception for the particular type of LLM usage. (***) shows significance of differences (p-values from regressions). Brackets indicate significance across all types of usage, whereas lines between squares indicate significance only for certain tasks.

We asked participants about their perceptions of the benefits and risks of LLM usage for research, the acceptability of such usage, and their comfort level with disclosing LLM usage to peers and reviewers, for each task category through Likert ratings. We also asked them to elaborate via free-response questions. In columns 2–6 of Figure 3, we present the average

---

[3]Here and in following results, we report $\mu$ as the mean of Likert ratings (min 1, max 5), and $p$ are p-values from post-hoc tests for significant pairwise differences among demographic groups using emmeans in Section 3.2.

rating given by participants, broken down by tasks and participants' backgrounds. Full significance tests can be found in Table 6 in the Appendix.

### 4.2.1 Perceptions of LLM benefits

Participants perceived greater benefits for Information Seeking ($\mu = 3.2$) and Editing ($\mu = 3.4$) compared to the other tasks ($\mu \leq 2.6$). Overall, we found **groups traditionally disadvantaged in the research community perceived more benefits**. Non-white researchers perceived more benefits than White researchers ($p < .0001$), as did researchers with 0–3 years of experience perceived more benefits than those with 11+ years of experience ($p = .0004$). Connecting findings on usage frequency and the correlation between perceptions and usage, together they suggest perceived utility drives higher LLM usage among non-white and junior researchers. Similarly, non-native English speakers perceived greater benefits in using LLMs for Editing ($p = 0.0004$), as well as reported using LLMs for Editing more frequently.

We also identified both **language equity** and **other equity** as major themes in respondents' discussions of LLM benefits (*"For honest researchers in resource-constrained developing countries, with little to no research funding, availability and use of LLMs is a game-changer leveling the playing field with other researchers in more fortunate climes."*). Benefits were especially highlighted by researchers facing systemic barriers, such as non-native English speakers, junior scholars, and researchers without much programming experience (*"I am not a native English speaker, so LLMs help me with the language barrier."*). We list all themes in Table 8 of the Appendix.

### 4.2.2 Perceptions of LLM risks and ethical concerns

To distinguish risks from ethical concerns, we asked participants to separately rate their perception of risks, given known issues with LLMs today, and their perception of the acceptability of using LLMs given a future where LLMs can prevent hallucinations and can always attribute any copyrighted text (if generated) to the original sources. Overall, researchers perceived using LLMs for Editing as not risky ($\mu = 2.5$), Direct Writing as moderately risky ($\mu = 3$), and the remaining categories as very to extremely risky ($\mu \geq 3.2$). Non-white researchers perceived fewer risks in using LLMs for Data Cleaning & Analysis ($p < .0001$), and Data Generation ($p = 0.0352$).

But in their acceptability ratings, researchers reported Ideation & Framing ($\mu = 2.9$) and Data Cleaning & Analysis ($\mu = 2.96$) as more unacceptable, while the remaining categories were found more acceptable ($\mu \geq 3.4$). Across all tasks, a researcher's **gender was significantly associated with their perception of the ethics of using LLMs**. Researchers who identified as women, non-binary, and other genders perceived LLM usage in research as less acceptable than those who identified as men ($p = 0.0017$). Similarly, **senior researchers** (11+ years of experience) **perceived less acceptability** than junior researchers (4–10 years of experience) for Editing ($p = 0.0215$) and Data Cleaning & Analysis ($p = 0.0013$). In contrast, non-white researchers perceived fewer ethical concerns for the more unusual usage categories, such as Data Generation ($p = 0.0230$), in keeping with their lower perceptions of risks, higher perceptions of benefits, and higher usage. Finally, computer scientists perceived using LLMs for Editing as more acceptable than social scientists ($p = 0.0008$), and they perceived Data Generation as more acceptable than natural scientists ($p = 0.0075$).

Participants used **strong language to share their opinions about the risks and ethics of LLMs for research** in their free-text responses (*"LLMs are tools for automated plagiarism and data fabrication that pose an existential threat to the network of trust essential for the integrity of academic work and the proper attribution of credit"*). While many pointed to specific risks like data fabrication and plagiarism, others drew attention to higher level concerns that could affect all of academic research, such as pollution of the research ecosystem with low-quality work (*"We need better judgment, slower science, and more thoughtful and ambitious work right now, not the opposite. Otherwise, we risk ridding science of its most special attributes just to crank out more papers."*). Respondents worried about future generations of researchers whose skills, diligence, and creativity may be impacted by over-reliance on LLMs (*"The main general risk is to flatten on 'average', which is the worst thing that may happen for a researcher...since this would block innovation"*). They also worried about exacerbating existing problems, like

overwhelming numbers of papers needing review (*"I fear for a deluge of AI-'assisted' (in the best case) papers that read somewhat fluently but are shallow, unoriginal, uninteresting, wrong in the details. This will overwhelm the peer review system"*). For a detailed overview of these themes, drawn from our qualitative analysis, see Table 9 in the Appendix.

### 4.2.3 Comfort with disclosure

We asked participants to rate their comfort with disclosing their use of LLMs for each of the six tasks. Overall, participants were **comfortable with disclosing to both peers and reviewers across all usage types** (disclose to peers: $\mu \geq 3.4$, disclose to reviewers: $\mu \geq 3.2$). However, we also found that **computer scientists reported more comfort with disclosure** than social scientists (to peers: $p = 0.002$; to reviewers: $p = 0.0043$) and biologists & medical researchers (to peers: $p = 0.001$; to reviewers: $p = 0.007$).

Qualitatively, participants' opinions on the disclosure of LLM usage and proper attribution were varied. One respondent mentioned *"academic shame"* as a reason to not disclose LLM usage. But other respondents highlighted the costs to the research community of not disclosing LLM usage: *"[If researchers don't disclose using LLM-generated text], I fear that researchers can get lazy, and we start having a lot of 'repeated text' in articles... and eventually researchers may just ask LLMs to generate the whole paper."* Some respondents listed this as their main concern with LLM-assisted research, though as long as LLM usage was disclosed, many described that usage as acceptable: *"The same sort of disclosure of use [as with human assistance] should be sufficient. The same responsibility for the integrity of work applies whether part of the effort was provided by a human assistant or an LLM."* Finally, one respondent called for better processes to support disclosure: *"...universities have totally different policies. It would be good if there was a generic system of how to indicate that editing or drafting tools were used."*

### 4.3 Does a researcher's usage of LLMs relate to their perceptions?

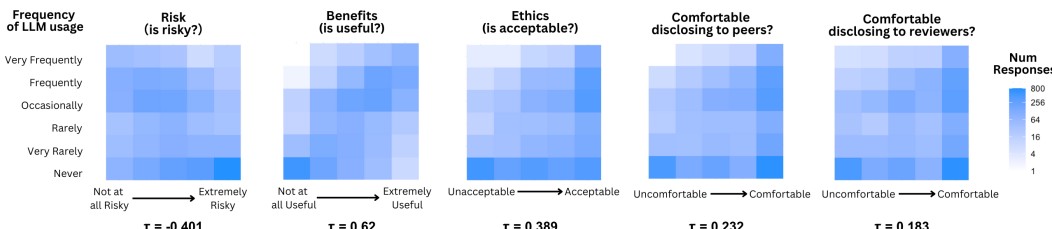

Figure 4: Each heatmap is for one type of perception, and each cell represents the number of responses (log scaled) that fall under this level of frequency of perception. The Kendall's tau coefficient on the bottom indicates how strong the correlation is between the usage frequency and the perception of that usage. All perceptions are significantly correlated with usage ($p < .0001$). Tests performed using `cor.test` in R and corrected with `p.adjust`.

Participants' perceptions of risks, benefits, and ethics and their willingness to disclose were all significantly associated with their usage of LLMs ($p < .0001$). As expected, greater perceived risks and ethical concerns were associated with lower usage, and greater perceived benefits were associated with higher usage (Figure 4). However, **perceived benefits had the strongest correlation to usage** ($\tau = 0.62$, $p < .0001$), and a weaker correlation for risk ($\tau = -0.401$, $p < .0001$) and ethics ($\tau = 0.389$, $p < .0001$, see Table 5 in Appendix). Some who perceived few risks or ethical concerns still reported infrequent usage. We also found a weaker but positive relationship between comfort with disclosure and frequency of LLM usage, with **discomfort highest for those who never used LLMs** for a given usage category. This suggests a possible lack of social norms around usage and disclosure may be a hindrance to adoption.

### 4.4 Do researchers care about the organizational governance behind LLMs?

Participants were split on whether the source of an LLM, (i.e., non-profit versus for-profit entities), impacted their perception of benefits and risks. 54.81% of participants (359) reported their perception would change depending on the model source while 45.19% (296) reported no difference. In their free-text elaboration, **59.07% (228) of respondents preferred LLMs from open source/non-profit entities,**[4] **while only 2.85% (11) stated they preferred LLMs from for-profit corporations**, and 38.08% (147) did not express a preference.

Qualitatively, the top reasons participants gave for their non-profit preference included the **incentives** of the organization, the **transparency** of the model, and **ethics and bias** considerations. These participants were skeptical of the incentives of commercial corporations, and worried that they would "*exploit user input, manipulate LLM outputs for financial gain.*" They also expressed concerns about monopolization and injecting bias to maximize profits. They favored non-profit entities because of the transparency in open-source models, increasing accountability, and users' trust. For the few participants who favored LLMs from for-profit entities, they believed those models were of higher **quality** due to the resources available to companies and their responsibility towards supporting customer issues. For participants who were indifferent, some held an attitude of **neutrality**: "*the technology is the same*" no matter which organization provided it. Some expressed that they cared more about the **quality** of the model, and would use the model with the best quality regardless of its source. Finally, other participants questioned the boundary between for-profit and non-profit entities: "*as we have seen with OpenAI, non-profits can easily become commercial.*" Some respondents prioritized whether the model was open source over whether it was developed by a non-profit or for-profit entity. Other respondents were skeptical about the open source label: "*No LLM is really open source. Most of them owe their existence to big commercial corporations, and even if they share the weights, we don't really know all the details about the training data. They are essentially black boxes.*"

## 5 Discussion

**Deep and Pervasive Integration of LLMs in Research** Our work revealed that most researchers have already found benefits in incorporating LLM-based tools into their current workflow, from literature review to data analysis. This confirms and expands upon prior assumptions about the prevalence of LLM usage in academia (Morris, 2023; Gruda, 2024; Bail, 2024; Koller et al., 2023; Kobak et al., 2024, summarized in §1). Participants in the free-form responses describe LLM tools with varying levels of autonomy and agency, ranging from direct manipulation ("*just another tool in the toolbox*") to data sources ("*a custom Wikipedia page*") to human-AI teaming ("*a useful research collaborator or assistant,*") to fully autonomous agents ("*an end-to-end AI researcher*"), which points to a wide design space of future LLM-based research support tools and user interfaces (Lee et al., 2024a). As researchers across AI and HCI domains continue to devote resources to developing new tooling, we may see increasing benefits in adopting LLM-based research support tools and a potential paradigm shift in the future of scientific work.

***"A Game-changer Leveling the Field":*** **Equity Benefits of LLMs** Our survey reveals compelling insights into how LLMs reshape research **equity** across demographic groups, a recurring theme of **equity** (§3.3, Appendix C) as a primary benefit of LLM use in research. Non-native English speakers described how LLMs allowed them to "*level the playing field*" by cutting down "*tedious and time-consuming editing tasks*" to "*more freely and precisely express ideas in another language [English].*" For example, §4.2 demonstrates that traditionally underrepresented or disadvantaged groups in research (Linxen et al., 2021)—specifically non-white researchers, junior scholars, and non-native English speakers—not only perceive greater benefits from LLMs but also report higher usage frequencies for certain tasks. Additionally, equity was mentioned in contexts such as enabling researchers without programming training to generate code for data cleaning, improving understanding of papers

---

[4]In the free response, some participants used open source and non-profit interchangeably. Thus, for the sake of labeling, we created a higher-level label of open source/non-profit to capture that opinion.

that are technical or from less familiar fields, or reducing monetary costs of proofreading services. This suggests that LLMs could help them overcome systemic barriers—including neo-colonial dynamics in research (Goyes and Skilbrei, 2023). To further build upon the equity benefits, future work should examine the unique needs of traditionally under-resourced groups for development of LLM-based research tools; for instance, future tools could educate junior researchers about research skills or practice critical thinking (Ye et al., 2024). However, the equity benefits are under scrutiny: while LLMs supported non-native English speakers in overcoming language barriers, the distinct "ChatGPT style" can be used to infer author demographics, potentially to their detriment (Lepp and Smith, 2025).

**Gender Gaps in LLM Usage and Ethical Concerns**   Despite some equity benefits, §4.2 indicates that women and non-binary participants expressed heightened ethical concerns regarding LLM use and demonstrated lower usage rates, though this difference did not reach statistical significance. This also extends beyond the research field: on public perceptions of AI, people who are nonbinary, transgender, and/or women reported significantly more negative AI attitudes compared to the majority groups, signaling that current AI tools fail to address their needs and concerns (Haimson et al., 2025). This pattern merits particular attention, given that these groups have historically faced disadvantages in academia. A concerning implication emerges: if ethical reservations lead these researchers to limit their LLM use, they might forfeit potential benefits and collaborative opportunities, potentially exacerbating existing inequities rather than alleviating them. The gender gap in LLM adoption has already been observed, where women and lower-earning workers were found less likely to use ChatGPT (Draxler et al., 2023; Humlum and Vestergaard, 2025). Future work should examine more deeply the specific ethical and other concerns expressed by these groups and strive to address their concerns in future LLM-based research tools. For instance, given prior research showing women scientists often struggle to receive appropriate credit for their work (Ni et al., 2021; Ross et al., 2022), their hesitancy about LLM use and attribution requires careful consideration.

**The Costs of Commercial LLMs in Research**   Our survey reveals that while commercial models are sometimes perceived to offer higher quality results and better user support, they also raise concerns about transparency, reproducibility, and more fundamentally, the misalignment of incentives between commercial entities and the research community. Many researchers have argued that the use of open-source models enhances the validity and integrity of research by allowing for greater scrutiny of research data and output (Sallou et al., 2024). These concerns should be carefully examined in high-stakes research areas such as medicine, bioengineering, and law, which can have direct, real-world impacts (Toma et al., 2023). In these fields, reproducibility and transparency are paramount, as they ensure the reliability of findings and provide justification for decisions that affect people's lives.

**Emerging Standards**   Our survey reveals concerns about the risks associated with LLM use in research, such as deskilling, decreasing creativity, and decreasing diligence (§3.3, Appendix C), which might result in the proliferation of low-quality research (Bail, 2024). Interestingly, our survey shows that while disadvantaged groups are more likely to discuss LLM *benefits*, perceptions of *risks* appear to be more uniformly distributed across demographics with few significant differences. For example, the problem of hallucinations in LLM outputs appears to be an equally significant concern for all researchers. This shared understanding indicates a collective awareness of LLMs' limitations and suggests the possibility of developing uniform standards for LLM use that can be broadly agreed upon, irrespective of people's demographic characteristics. The academic community has begun exploring various mitigation strategies such as a peer-reviewers' checklist (Watkins, 2023) and transparent disclosure of LLM use (Hosseini et al., 2023). Many conferences, publishers, and funding agencies have started to require LLM disclosure statements (ACL, 2024; ACM, 2024; AAAI, 2024; IEEE, 2024; 202, 2023; of Health, 2024; Kwong et al., 2024).

Our survey reveals that participants were broadly comfortable with disclosing LLM usage to both peers and reviewers across all tasks, though these varied by discipline, with computer scientists reporting significantly higher comfort compared to researchers in social sciences & humanities or biology & medicine. This disparity reflects that while some disciplines may

accept LLMs as standard research tools, others are still grappling with integrating these technologies into their established practices. The mention of "*academic shame*" by one respondent highlights the ongoing stigmatization of LLM use in some research communities. The lack of standardized policies across institutions further complicates this landscape, creating additional burdens for individual researchers who must navigate varying expectations and norms. This suggests that different academic communities are still negotiating their norms of acceptable LLM use, and continued discussion is crucial to fully leverage LLMs towards improving science.

## 5.1 Limitations

While our recruitment method covers a wide range of fields, more of our participants were computer scientists (40%), though other fields such as social sciences, biology, medicine, and natural sciences were also represented. There were also more men who responded to the survey (79%), which may partly reflect existing imbalances. The detailed distributions are reported in §4 and Appendix A.3. Our data also includes: (1) a high proportion of more senior white researchers; (2) more researchers who identified as men in the natural science & engineering field, and (3) fewer senior researchers in the computer science field. These could be the result of sampling bias or existing imbalances in these respective fields.

The survey responses were collected in batches of recruitment emails over a six-month period from November 2023 to April 2024. The uses and perceptions of researchers may change as LLM tools continue to evolve, but we hope this survey can give readers a snapshot of the current state of the community and a baseline data point in time for comparisons done by future studies, together supporting informed decisions as we continue to build consensus and norms around the use of LLMs for research.

Our quantitative analysis assumes that the 5-point rating data follow an unimodal distribution, which supports the use of parametric statistical methods. While this approach is acceptable (Harpe, 2015), we acknowledge that multimodal distributions could challenge the validity of these methods.

Finally, we discuss the limitations of our regression method. We chose to fit separate regression models for each demographic variable—one model to analyze race on perception and a separate model to analyze gender on perception—as a way to improve statistical power and interpretability of results. Fitting a more complex model like $\text{Rating}_{ij} \sim \text{Race}_i * \text{Gender}_i * \text{Year}_i * \cdots * \text{UsageType}_{ij}$ would lack degrees of freedom impeding statistical testing, as well as result in a larger regression model with over 500 estimated coefficients. Such models would be difficult to interpret and require making additional assumptions about the modeled relationships between demographic variables (e.g., are race and gender linearly related in their effects on perception). Our choice to fit individual models for each demographic variable eschews these problems, though does not capture interactions between pairs of demographic variables. We provide tests of independence between demographic pairs in Appendix Table 2, where we observe correlations between three pairs (Gender & Field, Race & Year, Year & Field).

## 6 Conclusion

In this study, we ran a large-scale survey of diverse groups of researchers about their usage and perception of LLMs for research, and revealed the widespread adoption of LLMs and distinct perceptions based on academic disciplines and demographics. Our work suggests that the research community is at a critical juncture, balancing growing LLM integration with the need to uphold originality, rigor, and ethical conduct as well as the potential for a more productive and inclusive global research landscape. It also underscores the need to better understand the implications of LLM usage—not just technical integration but also sociological, ethical, and epistemological impacts across disciplines and researcher demographics. We call for studies that examine long-term effects of LLM use on research quality, creativity, and the development of research skills as well as investigations into the potential of LLMs to increase fairness and representation in academia.

## 7 Ethics Statement

We recruited from verified authors listed on Semantic Scholar that has a dedicated data quality team maintaining academic author profiles. We could have tied the survey responses to participants' author metadata from Semantic Scholar to obtain high-precision demographic information (such as a list of publications, years of experiences, institutions, pronouns, etc.). However, for privacy concerns, we only used their email addresses for targeted recruitment of verified published authors. We instead relied on self-reporting using optional survey questions for demographic information, and did not tie survey responses to their author metadata.

We surveyed participants about their racial or ethnic identity. Some participants who chose to self-describe their identities highlighted limitations in the provided options, such as including only one generalized "Asian" category, while others chose not to respond because they found the options to be U.S.-centric (see all survey questions in the supplementary material). We acknowledge these limitations in the design of this question and note the diversity of our participants, with 24% self-reporting U.S., the most common country of origin among respondents (see Appendix A.3.2 about other top countries).

The authors would like to thank our many anonymous survey respondents who provided long and thoughtful opinions and insights in the optional free-text survey questions. These responses enabled our qualitative analysis and showcased that many researchers are actively contemplating and engaging in conversations around the use of LLMs as a research support tool today.

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

# A Methods

## A.1 Survey Design

When designing the questionnaire in the survey, we used the four following approaches: First, for inspiration, we looked to recent literature on using LLMs as a productivity tool for research (Morris, 2023; Messeri and Crockett, 2024; Bail, 2024; Russo et al., 2024; Wang et al., 2023) and other scenarios (Liang et al., 2024a; Liu et al., 2022), which included qualitative interviews and survey results. Second, we reviewed historic papers on how the research community had adopted new tools in the past, specifically around the use of crowdsourcing for data collection, user studies, and other productivity tasks (Law et al., 2017; Kittur et al., 2008; 2013). Third, we publicized an anonymous formative survey on X/Twitter targeted towards researchers with open-ended questions about whether and how they use LLMs for research in order to help define initial categories of usage that we later refined. Finally, we shared early drafts of the questionnaire with other researchers in our own institutions for feedback and iteration. Through this process, we classified LLM usage for research into six broad categories: information seeking, editing, ideation & framing, directing writing, data cleaning & analysis, and data generation.

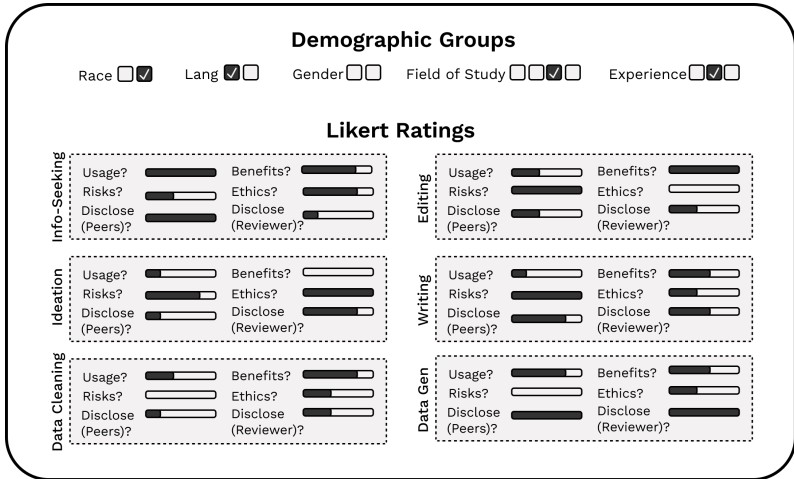

Figure 5: An example of data we collected per participant from sections of the survey that relate to our statistical modeling. Each participant provides answers to (up to) five demographic questions and (up to) 36 Likert ratings in response to questions about LLM usage frequency, perceptions, and usage types. Participants are not required to report on every question. In this example, the gender information is missing from the participant.

## A.2 Recruitment

A recruitment email was sent to 100,187 verified authors, using a Semantic Scholar email list compiled over the past decade, although the email addresses were not guaranteed to be active. Out of all sent emails, 34,922 emails were opened (34.9% open rate) and 1,834 emails were clicked on (1.8% click-through rate). After click-through, 1,226 participants signed the consent form to start the survey, and thus the survey response rate lies between 1.2% to 3.5%, depending on what proportion of our targeted email addresses were no longer active. Finally, after filtering out survey results that did not progress past the first page or spent fewer than 2 seconds on each question, we ended up with 816 total responses.

## A.3 Participants Demographics and Quantitative Method

We transformed the dataset to generate the final demographic groups in which some of the response options were grouped to form coarser buckets (e.g., years of research experience),

and free responses (e.g., fields of study) were manually coded and discretized for analysis. Answers that did not fit into any categories, such as "Prefer Not to Answer," were filtered out from demographic-specific analysis. We ended up with 611 responses with gender identity, 527 with racial identity, 644 with years of research experience and native language information, and 635 with field of study information.

|  | Man | Woman, Non-Binary, Other |
|---|---|---|
| White | 243 | 74 |
| Non-White | 168 | 34 |

$p = 0.7578$

(a) Race and Gender

|  | 0-3 | 4-10 | 11+ |
|---|---|---|---|
| White | 27 | 96 | 198 |
| Non-White | 31 | 81 | 94 |

$p = \mathbf{0.0101}$

(b) Race and Years of Experience

|  | CS | Bio | Nat.Sci | Soc.Sci |
|---|---|---|---|---|
| Man | 194 | 65 | 113 | 110 |
| Woman, Non-Binary, Other | 48 | 22 | 12 | 40 |

$p = \mathbf{0.0008}$

(c) Gender and Years of Experience

|  | CS | Bio | Nat.Sci | Soc.Sci |
|---|---|---|---|---|
| 0-3 | 37 | 6 | 11 | 12 |
| 4-10 | 102 | 30 | 39 | 35 |
| 11+ | 118 | 58 | 82 | 105 |

$p = \mathbf{0.0030}$

(d) Years of Experience and Field of Study

Table 1: Contingency tables of participant counts for different demographic pairs. $p$-values from Chi-square tests of independence; lower values are interpreted as evidence for dependence between variables.

### A.3.1  Consolidating Demographics Groups

To balance our analysis given a large proportion of men participants, we collapsed all responses from women, non-binary, and other participants into a single category. Similarly, we collapsed participants identified as Asian, Black, Hispanic or Latino, and Middle Eastern into a single category of non-white because of a large proportion of white participants. See detailed racial categories below. Field of study was collected as free response and manually classified into four categories by the authors. We collected the research fields that our participants studied in the form of free responses. Out of the 816 responses, 644 (79%) responses included field information. We classified 635 free responses into four field categories: computer science, social science & humanities, natural science & engineering, and biology & medicine, and 9 responses were classified as other and excluded from the analysis. Computer science group had 257 participants and also included interdisciplinary fields with computer science, such as education technology, except biotechnology. Social science & humanities had 152 participants and included psychology, behavioral science, education, sociology, and more. Natural science & engineering had 132 participants, and included math, chemistry, physics, environmental science, electrical engineering, and more. Biology & medicine had 94 responses, and included cognitive science, bioinformatics, biotechnology, public health, neuroscience, and more. We included interdisciplinary fields between biology and computer science in the biology & medicine group instead of the computer science group to balance the number of participants in each field.

After consolidating the demographic groups, we inspected the correlation between certain demographic groups: a series of Chi-square tests of independence in R (`chisq.test`) between all pairs of the five demographic groups, with multiple comparisons $p$-value correction using Holm-Bonferroni (`p.adjust`). Through a series of Chi-square tests and Holm-Bonferroni correction, we found that most variables appear independent, except for three pairs with significant $p$-values: race and years of experience, gender and field of study, and years of experience and field of study.

|          | Gender | Race   | Year   | Publication | Field  |
|----------|--------|--------|--------|-------------|--------|
| Gender   | /      | 0.7578 | 1.0000 | 0.1433      | **0.0008** |
| Race     |        | /      | **0.0101** | 1.0000  | 1.0000 |
| Year     |        |        | /      | 1.0000      | **0.0030** |
| Language |        |        |        | /           | 1.0000 |
| Field    |        |        |        |             | /      |

Table 2: $p$-values from Chisquare tests of independence between demographic groups. All $p$-values were corrected via Holm-Bonferroni. Significant values ($p < 0.05$) are in **bold** and indicate dependence between pairs of variables. We previously also had another variable representing researcher experience—number of publications—but found that it was highly correlated with Years of research experience (Chisq Independence test; p-value = 3.3E-15).

### A.3.2 Participants' Country of Origin

| Country | Percentage | Number of Participants |
|---------|------------|------------------------|
| United States of America | 23.80% | 154 |
| Germany | 6.34% | 41 |
| Canada | 4.33% | 28 |
| Italy | 4.17% | 27 |
| India | 4.17% | 27 |
| United Kingdom of Great Britain and Northern Ireland | 3.25% | 21 |
| Spain | 2.47% | 16 |
| China | 2.47% | 16 |
| Australia | 2.47% | 16 |
| France | 2.32% | 15 |
| Nigeria | 2.16% | 14 |
| Brazil | 2.01% | 13 |
| Sweden | 1.85% | 12 |
| Russian Federation | 1.85% | 12 |
| Netherlands | 1.85% | 12 |

Table 3: Top 15 countries of origin reported by participants

### A.3.3 Participants' Racial Identities

Among all participants who provided their racial identities, 51.4% identified as white/Caucasian, 17.8% identified as Asian, 3.6% identified as Black or African American, 8.1% identified as Hispanic or Latino, 2% identified as Middle Eastern. Additionally, 8.9% selected Prefer not to disclose and 8.1% selected Prefer not to disclose, which were excluded from the demographic analysis.

### A.4 Quantitative Methods

In practice, in addition to the regression model described in the main text, we always attempt a second model fit that includes an *interaction term*—$\beta_3 \texttt{Demographic}_i \times \texttt{UsageType}_{ij}$. We conduct a likelihood ratio test using `anova()` in R between each model against a *null* model which has no demographic variable (i.e., the null hypothesis of no demographic effects) and choose the interaction model if the interaction term is statistically significant; otherwise, we defer to the simpler model without an interaction term. In our analysis, we find that rarely is the interaction term needed.

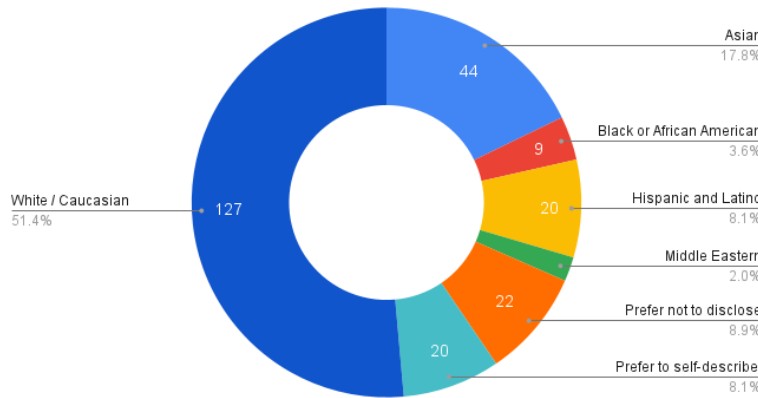

Figure 6: Among all participants who provided their racial identities, 51.4% identified as white/Caucasian, 17.8% identified as Asian, 3.6% identified as Black or African American, 8.1% identified as Hispanic or Latino, 2% identified as Middle Eastern.

## B Quantitative Results

|                              | Gender | Race | Language | Years | Field | All  |
|------------------------------|--------|------|----------|-------|-------|------|
| RQ2  Frequency               | 3666   | 3162 | 3864     | 3864  | 3810  | 4896 |
| RQ5.1 Risk                   | 2973   | 2597 | 3110     | 3110  | 3066  | 3410 |
| RQ5.2 Benefits               | 2315   | 2016 | 2440     | 2440  | 2403  | 2738 |
| RQ5 3 Ethics                 | 3191   | 2744 | 3354     | 3354  | 3314  | 3665 |
| RQ5.4 Disclosure to Peers    | 3061   | 2631 | 3207     | 3207  | 3168  | 3493 |
| RQ5.4 Disclosure to Reviewers| 3028   | 2637 | 3177     | 3177  | 3139  | 3450 |

Table 4: Number of answers to survey questions across all participants, broken out by demographic and question. Each question (row) had six sub-questions; participants did not have to answer all questions.

| Type of Usage          | Risk   | Benefit | Ethics | Disclosure to Peers | Disclosure to Reviewers |
|------------------------|--------|---------|--------|---------------------|-------------------------|
| Information Seeking     | -0.303 | 0.513   | 0.261  | 0.127 (0.0005)      | 0.078 (0.041)           |
| Editing                | -0.276 | 0.557   | 0.282  | 0.171               | 0.073 (0.041)           |
| Ideation & Framing      | -0.372 | 0.651   | 0.385  | 0.197               | 0.177                   |
| Direct Writing          | -0.401 | 0.575   | 0.409  | 0.184               | 0.136 (0.0004)          |
| Data Cleaning & Analysis| -0.437 | 0.68    | 0.335  | 0.229               | 0.186                   |
| Data Generation         | -0.45  | 0.665   | 0.383  | 0.262               | 0.245                   |
| Overall Usage           | -0.401 | 0.62    | 0.389  | 0.232               | 0.183                   |

Table 5: Kendall's tau correlation between the frequency of LLM usage and the perception of that usage. Each cell includes the tau coefficient with the Holm-Bonferroni corrected $p$-value in parenthesis. Coefficients not followed by parenthesis all had $p<.0001$.

| | Man-(Woman, NonBin., Oth.) | $p$ | NonWhite-White | $p$ | 11+ yrs-0-3 yrs | $p$ |
|---|---|---|---|---|---|---|
| Benefit (1-5) | | | 0.420 | $< .0001$ | -0.4187 | 0.0004 |
| Ethics (1-5) | 0.351 | 0.0017 | | | | |

| | Bio-CS | $p$ | CS-Soc.Sci | $p$ |
|---|---|---|---|---|
| Disclos.Peers (1-5) | -0.644 | 0.0001 | 0.513 | 0.0002 |
| Disclos.Reviewers (1-5) | -0.622 | 0.0007 | 0.451 | 0.0043 |

Table 6: Post-hoc tests for significant pairwise differences in LLM perception ratings between demographic groups across all LLM usage types. This table reports the rating differences between demographic levels and associated $p$-values from emmeans. For example, in the column for race and the row for Benefit, the result shows that Non-White researchers, on average, report 0.42 points higher ratings than White researchers on perceived benefits of LLM usage. Only statistically significant results are included.

## C  Qualitative Themes

| Theme | Description | Example |
|---|---|---|
| **Reproducibility** | Whether research is reproducible/replicable/verifiable by other researchers | *"Because research should be reproducible, and this is only possible with the use of open-source LLMs."* 
 *"...an open-source model/data could be used and verified by external researchers and, hence, be more trustworthy."* |
| **Transparency** | Whether the model release is transparent, including code, training dataset, evaluation dataset, etc. | *"Greater transparency on the models and training data would increase my confidence in the models' accuracy."* 
 *"As researchers, it is not just sufficient to use the service as a black box. I would like to know more about how the models were trained and what data went into it. It builds trust and expands the knowledge."* |
| **Availability** | Whether the model is widely available, released to the public, and has no/low barriers for researchers to access | *"World don't have similar access to AI that might result in systemic discrimination."* 
 *"...Using for-profit solutions produces costs that require funding, but open-source options are typically less 'ready-to-use' and often require setup and potentially also local/available computing resources."* |
| **Accountability** | Who should be responsible/accountable for the model and its use | *"I don't think it makes a difference whether I'm using os tech or ChatGPT, it's still my responsibility to diligently check the outputs and the responsibility for any risks is on me as the user."* 
 *"A paid, close-source product could be less reliable, but there is someone to sue if things go awry."* |
| **Privacy** | Whether data is kept private and stored securely | *"Open-source entities are more reliable, and transparent. The risk of using copyrighted data is less. The risk of stealing my personal sensitive data is less."* 
 *"I trust open-source software more than proprietary/commercial corp if I have to handle personal/sensitive data."* |
| **Incentives** | What kinds of incentives drive the creation and release of the model | *"LLMs are fundamentally dependent on using people's actual creative and artistic work without their consent. The motives of the LLM's creator have no effect on that. Motives similarly have no effect on LLMs' unreliability. Motives do not change whether LLMs are accountable for their mistakes, because LLMs cannot be accountable."* 
 *"I can't trust the objectives and implementations of a closed commercial corporation."* |
| **Neutrality** | Whether tools are considered neutral and who created/owns the tool does not matter | *"The technology is the same (whether it's provided by a commercial or an open source entity), so my perceptions are the same."* 
 *"I don't know enough about what's going on (or not) 'behind the scenes' in LLMs regardless of where they're from—my perception of them remains that they're unethical and not useful."* |
| **Ethics & Bias** | Whether the model is biased or uses unethical methods and data | *"...a for-profit company might produce models which are favouring a certain kind of opinion, line of thought, or products. These might be equivalently good/bad to a non-profit model, but poses this inherent bias by whomever is currently financially invested into the entity"* 
 *"The fundamental issues of bias, hallucination, ethical issues of invention, the need for review, etc. will not change whether the LLM is open-source or closed."* |
| **Quality** | Whether the model has good performance (outputs are useful and of high quality) | *"I´d rather use open-source LLMs, but I acknowledge that their performance is still behind commercial models which makes it difficult to use them."* 
 *"I would be concerned that the commercial aspects would affect the results served. Perhaps we would be directed more to paid sources."* |

Table 7: Themes found in **Q64:** *Would your perceptions of the benefits and risks of using LLMs be affected by whether the LLM is part of **an open-source or non-profit entity versus a commercial corporation**? Why or why not?*

| Theme | Description | Example |
|---|---|---|
| **Language Equity** | Bridging language gaps to support non-native English speakers | *"I am not a native English speaker, so LLMs help me with the language barrier. "* 
 *"Because I'm not Native American, I've received a number of negative comments from reviewers, and I've always wondered how I can write like a Native American if I'm not one. Today, with the LLM tools, I can understand the terms I use that aren't Native American, and I'm able to improve my writing."* |
| **Other Equity** | Removing barriers for researchers (not language), such as supporting neuro-divergent researchers, researchers with limited resources, or researchers without programming experience | *"For researchers without programming skills, LLMs allow data analysis without programming. What they need to learn is how to make good natural language prompts."* 
 *"Improve understandability for non-specialists"* 
 *"For honest researchers in resource-constrained developing countries, with little to no research funding, availability and use of LLMs is a game-changer leveling the playing field with other researchers in more fortunate climes. "* |
| **Efficiency** | Saving time and resources during the research process | *"Enhancing work efficiency and the capability of information gathering and extraction, thereby enabling researchers to focus more energy on creative tasks."* 
 *"Speeding up the academic writing/reading will speed up the research cycle of the community."* |
| **Routine Task Assistance** | Completing routine or repetitive tasks, freeing the researcher to focus on higher level tasks | *"I think the main benefit could be saving time that would otherwise have to be spent on relatively 'mechanical' tasks, such as literature search, writing code for data cleaning and analysis, creating nice figures for publication and similar."* 
 *"Anything repetitive would benefit from LLMs"* |
| **Search** | Search and information retrieval tasks, which might include literature review | *"Information seeking actions (like search) seem to often depend on serial reformulations of concepts, rephrasings, etc. to try to find the right thing. Even if LLMs hallucinate sources, this can be very helpful in those intermediary steps that are part of searching for things. There's just too much out there."* 
 *"LLMs can give you enough information to continue searching online."* |
| **Literature Review** | Helping with literature-related tasks, such as finding related work, writing literature reviews, and summarizing literature | *"Easing the otherwise time-consuming and burdensome processes in research. Tasks like lit searching, reading through content to determine study strengths/weaknesses/biases, extracting data from pubs..."* 
 *"The internet is one big pile of noise. Publications are a somewhat less noisy, but still rather noisy pile of information. LLM's cut through that to some degree."* |
| **Editing** | Editing tasks, such as rephrasing sentences | *"Mostly rephrasing, rewriting, condensation, bulleting"* 
 *"Editing and language perfection, something like an advanced Grammarly."* |
| **Overcoming Writer's Block** | Helping researchers to begin writing (to put something on the blank page) | *"The major benefit of using LLMs comes in action when we are stuck, for example not knowing the exact word or term, or not finding the answer to some of the ideas about how it can be used or how it can be applied."* 
 *"...loosing the fear of a blank sheet of paper when you don't know how to start your article..."* |
| **Broadening Perspectives** | Helping researchers discover perspectives and diversify their sources | *"LLM are remarkable important as they reduce generic data and improve novelty of work"* 
 *"Always available "colleague" to discuss ideas with and get feeback from."* |
| **Programming** | Supporting programming tasks, including debugging and writing new code | *"I basically code using CoPilot + GPT now, often for research glue code."* 
 *"Definitely saves time on coding applications where I'm not aware of certain routines or packages (for example python packages that already exist). Very helpful to get syntax correctly. Get feedback on coding errors..."* |
| **Brainstorming** | Helping brainstorm, organize ideas, and get feedback on ideas | *"LLMs are a great tool to help you create hypotheses, as a way to brainstorm, where there really are no wrong ideas and therefore you cannot suffer with any potential misleading information, as you are expected to have domain expertise anyway."* 
 *"Having an always available writing companion to help with ideation, divergent thinking, encouragement, and general advice."* |

Table 8: Themes found in **Q65:** *Given your ratings above on the benefits/usefulness of these different ways of using LLMs, can you explain what you see to be the main **benefits/usefulness** for the academic community and for you as a researcher?*

| Theme | Description | Example |
|---|---|---|
| **Hallucination & Misinformation** | Production and spread of incorrect information invented by the model | *"Sometimes it creates so complicated hallucinations so that even an expert can think that what it writes it true although it is not."* 
 *"Putting more falsehoods into [the internet's] shared memory is a crime."* |
| **Inaccuracy** | Incorrect conclusions and analyses | *"There is a risk of less experienced scientists using these technologies as they are unable to check if the outputs are correct as easily as someone with more experience/intuition."* 
 *"The risks are proportional to prior knowledge of the subject."* |
| **Biases** | Model's outputs could contain biases and stereotypes | *"Promotions of some papers more than others - further marginalizing voices of those who are already less discovered and less cited despite similar quality of papers."* 
 *"I worry about biased LLMs influencing the research directions we choose and the conclusions we draw."* |
| **Lack of Disclosure** | Attribution or disclosure of LLM usage | *"Not acknowledging the use of the AI - universities have totally different policies."* 
 *"The advancement of this technology will make these models a kind of co-author, to the point where we will not know the real contribution of the human component."* |
| **Plagiarism** | Risks of plagiarism when using models to generate paper text | *"Blind trust in a system that is hard to understand which could lead to accidental plagiarism as it's not easy to understand which information LLMs base their outputs on."* 
 *"plagiarism at scale, the community doesn't have enough time to check all the existing papers"* |
| **Disrespecting Authorship** | Copyright and other concerns related to ownership of training data and model outputs | *"...its use to profit off of text whose authors weren't compensated is pretty fucked up."* 
 *"There are issues of integrity where the data that is trained on doesn't necessarily belong to the model makers, and the model's output doesn't belong to the user."* |
| **Fabrication** | Using LLMs to fabricate data and research results | *"The risk of reporting 'results' based on synthetic data without actually having conducted any experiment."* 
 *"LLMs are tools for automated plagiarism and data fabrication that pose an existential threat to the network of trust essential for the integrity of academic work and the proper attribution of credit."* |
| **Decreasing Creativity** | Effects of model use on the creativity and originality of research work | *"...llms are going to make people less creative over time. this needs of course more thinking and evidence, but to me, ppl may not start thinking or collaborating human to human to find valuable H2H collaborative ideas, but rather M2H ideas, which might miss the human touch."* 
 *"The main general risk is to flatten on "average", which is the worst thing that may happen for a researcher (and it is already happening for arts such music, since this would block innovation."* |
| **Pollution of Research Ecosystem** | Decreasing quality of research that leads to overall pollution of information and community | *"The huge number of poor quality papers out there are already making science more difficult, and I can see LLMs making this much worse."* 
 *"We need better judgment, slower science, and more thoughtful and ambitious work right now, not the opposite. Otherwise, we risk ridding science of its most special attributes just to crank out more papers."* |
| **Decreasing Diligence** | Over-reliance on and trust in models leading to decreasing diligence of researchers | *"Speed, copy/paste attitude, less research propositions, less reading of the full article, not enough training..."* 
 *"The main risks are related to human tendency to be lazy and appreciate convenience too much. It would be easy to overtrust LLM output the inherent opacity of the technology invites people to do so. It is hard to check why particular output is produced and people have a natural talent to explain and justify things, even completely wrong things when it is more convenient."* |
| **Deskilling** | Loss of research skills due to reliance on models | *"...sort of like with self-driving cars which, if we ever get ones that work a bit, people will stop learning how to drive, which will be bad when the things actually get confused and hand you the wheel!"* 
 *"Learning how to relay information is an important skill that must be cultivated throughout ones career. The process of writing the information gives you the way to check whether what you have done actually makes sense and provides value."* |

Table 9: Themes found in **Q66:** *Given your ratings above on the risks/ethical considerations of these different ways of using LLMs, can you explain what you see to be the main* **risks/ethical considerations** *for the academic community and for you as a researcher?*

