# OpenReview forum: "LLMs as Research Tools: A Large Scale Survey of Researchers’ Usage and Perceptions"
_colmweb.org/COLM/2025/Conference — COLM 2025_

### Official Review · Reviewer_u4YT · 2025-04-29

**Rating:** 4
**Confidence:** 4
**Ethics Flag:** 1

**Summary:**

The study conducted a systematic survey of 816 researchers to quantify the current usage of large language models (LLMs) in scientific research. Results showed 81% of researchers have adopted LLMs, primarily for information retrieval and text editing, with less use for data analysis and generation. Findings revealed non-white, junior, and non-native English speaking researchers use LLMs more frequently and perceive greater benefits, while female and non-binary researchers expressed stronger ethical concerns. Computer science researchers showed highest acceptance, with overall preference for open-source/non-profit models. The research suggests LLMs may enhance research equity but require addressing ethical issues and establishing disciplinary norms. It provides the first empirical evidence on LLM adoption patterns and perception differences across demographic groups in academia.

**Reasons To Accept:**

1. Approximately 81% of surveyed researchers utilize LLMs, primarily for information retrieval and text editing tasks, while employing them less frequently for data analysis and content generation.

2. Non-white, junior, and non-native English speaking researchers demonstrate higher LLM adoption rates and more positive perceptions of benefits versus risks, whereas female and non-binary researchers report heightened ethical concerns.

3. Although LLM usage spans all disciplines, computer scientists exhibit greater transparency in usage reporting and fewer ethical reservations compared to other fields, with researchers overall favoring open-source/non-profit LLM platforms.

**Reasons To Reject:**

1. The paper lacks sufficient novelty and offers limited reference value.

2. While the study surveyed 816 researchers, it failed to adequately distinguish between different academic levels (e.g., senior scholars vs. junior researchers) and disciplines. In reality, researchers at different career stages and from various fields may have substantially different motivations for using LLMs.

---

> ### Author Response · Authors · 2025-06-03
>
> Thank you for your review. Our responses to your review are as follows:
> ## Lack of novelty and reference value of this work
> Compared to prior work that relied on small-scale interviews, our work used survey data that enabled us to understand how researchers leverage LLMs in their daily research **at a larger scale** compared to their perceptions and sentiments **across different demographic backgrounds**, none of which were covered by prior work.
>
> We disagree with the reviewer’s assessment of the reference value of this work. Researchers’ use of LLM as research tools and their perceptions is of **high interest across many research communities**. As our Related Work section points out, AI/LLM tools for research is a fastly evolving field with lots of attention both from the research field and commercial companies. Some anecdote examples are that Nature’s previous coverage on this topic, _[Scientific Discovery in the age of artificial intelligence (2023)](https://www.nature.com/articles/s41586-023-06221-2)_, has received 1200+ citations. And, a similar survey study, _[AI and science: what 1,600 researchers think (2023)](https://www.nature.com/articles/d41586-023-02980-0)_, also received 200+ citations. Our paper goes beyond these, not just looking at perception but also actual **usage** as well as analyzing across more different demographics. As LLM technologies improve rapidly, it is important for us to measure and understand usage and perception patterns at scale across these communities, which is the focus of our work.
>
> ## Failed to distinguish between academic levels and disciplines
> We disagree with the reviewer on this assessment. In fact, **the academic level and disciplines of researchers are central demographic information to our analysis**. We mentioned this in multiple places throughout the paper.
> - In the Abstract, _"Findings revealed non-white, **junior**, and non-native English speaking researchers use LLMs more frequently and perceive greater benefits… **Computer science researchers** showed highest acceptance, with overall preference for open-source/non-profit models”_
> - In the Method section (111-118),  _“Of the 816 total responses, 644 included demographic information, with the following distributions…**Years of Research Experience: 11+ (57%); 4-10 (32%); 0-3 (11%)**...**Field of Study: Computer Science (40%); Social Science & Humanities (24%); Natural Science & Engineering (21%); Biology & Medicine (15%).** ”_
> - Multiple places in Findings and Discussions, listing a few. _“Non-white researchers perceived more benefits than White researchers (p < .0001), as did **researchers with 0–3 years of experience** perceived more benefits than those with **11+ years of experience** (p = .0004).”_ (170 - 171) _“Similarly, **senior researchers (11+ years of experience)** perceived less acceptability than **junior researchers (4–10 years of experience)** for Editing (p = 0.0215) and Data Cleaning & Analysis (p = 0.0013).”_ (198-199) _“Finally, **computer scientists** perceived using LLMs for Editing as more acceptable than **social scientists** (p = 0.0008), and they perceived Data Generation as more acceptable than natural scientists (p = 0.0075).”_ (203-204)

---

> > ### Comment · Reviewer_u4YT · 2025-06-04
> >
> > 1. "Years of Research Experience" does not equate to academic levels—professional rank/title would provide clearer differentiation.
> >
> > 2. The categorization of "Field of Study" lacks logical consistency. Why is Computer Science not grouped under Natural Science & Engineering? Additionally, the current sample size for Computer Science appears disproportionately large.

---

> > > ### Author Response · Authors · 2025-06-05
> > >
> > > ## Years of research experience do not equate to academic levels
> > > We chose Years of Research Experience to approximate a researcher’s seniority because it is a more **objective and comparable measure across diverse affiliations**. Professional rank or title varies significantly across countries and institutions (e.g., academia vs. industry). While we agree that no single metric perfectly captures a researcher’s seniority, we believe Years of Research Experience is a consistent and practical measure, given the **large and diverse sample** in our study (75% from university, 8% from commercial company, 5% from NGOs, 6% from governmental institutions, and the rest from other instituions). Additionally, we believe that our findings are robust to alternative definitions of seniority; **a slightly different definition is unlikely to alter the survey results or their validity substantially**.
> > >
> > > ## The categorization choice for Field of Study
> > > We grouped Computer Science separately from the Natural Sciences & Engineering category because CS comprised a larger share of our sample (40%). Grouping CS with other fields would obscure meaningful differences and prevent adequate comparison across disciplines.
> > >
> > > This decision is consistent with our broader strategy to manage demographic imbalances. For instance, we collapsed women, non-binary, and other gender identities into a single group to compare against men; likewise, we aggregated multiple racial/ethnic categories into a broader “non-white” group for analysis. These decisions allowed us to **preserve statistical power and enabled analysis across groups** despite the sampling skew. We elaborate on the Field of Study grouping process in the Appendix, lines 627–637.
> > > ## Higher proportion of computer scientists among participants
> > >
> > > As all survey studies must grapple with the issue of self-selection bias, which would influence respondent demographics, we have mitigated this in several ways:
> > >
> > > First, **we controlled potential demographic skew in our statistical models**. We conducted subgroup analyses isolating each demographic (comparisons between CS and other fields, white versus non-white…), to ensure that our findings are not solely driven by the dominant group. Furthermore, several of our findings are specific to or vary across demographic groups, and our discussions are sensitive to and reflective of these differences.
> > >
> > > Second, **we are completely transparent about our process and results**. We share our survey questions, data, and the analysis code. Finally, we acknowledge the unbalanced demographics in our sample (line number 322 - 329).

---

> > > > ### Comment · Reviewer_u4YT · 2025-06-05
> > > >
> > > > The paper lists Computer Science separately instead of including it in the Natural Sciences & Engineering category. It is recommended to add robustness checks—specifically, by reclassifying Computer Science under Natural Sciences & Engineering and examining whether the results exhibit substantial or statistically significant changes.

---

### Official Review · Reviewer_5NYA · 2025-05-07

**Rating:** 7
**Confidence:** 4
**Ethics Flag:** 1

**Summary:**

This paper is among the first to conduct a large-scale survey on LLM usage as a research tool. The survey contains questions on self-reported demographics, LLM usage, and perceptions. The authors contacted 107,346 verified authors with a final response rate of 0.8%. They find that disadvantaged groups in academia report higher LLM usage and perceived benefits, while women, non-binary, and senior researchers have greater ethical concerns.

**Questions To Authors:**

See reasons to reject.

**Reasons To Accept:**

As the authors state, this is among the first studies that provides much-needed evidence about how LLMs are currently used by researchers. The authors made a great effort in gathering and processing this data, and it is nice to see that they have attached their data and code (I believe they will also open source it upon acceptance?).

**Reasons To Reject:**

Given the survey’s low response rate, selection and non-response biases are potential concerns that may overestimate actual LLM usage. Additionally, the high proportion of computer scientists among respondents (as acknowledged in the limitations) likely skews results toward disciplines already inclined toward technology adoption, affecting generalizability. It would strengthen the discussion if authors could clarify whether any anonymized metadata on non-respondents exists, despite understandable GDPR constraints, to assess how representative the sample is.

---

> ### Author Response · Authors · 2025-06-03
>
> Thank you for your review, and thank you for acknowledging that we are **”among the first studies that provides much-needed evidence”** on this topic, and recognizing that we’ve **”made a great effort in gathering and processing this data”**. Indeed, **we will open source the survey questions, results, and the analysis code** upon acceptance. Our responses to your review are as follows:
> ## Impact of survey study non-response and selection bias
> Selection and non-response bias are inherent to the survey methodology. We believe we have approached this in a manner that places our survey methodology among the most rigorous within this space. For example, let’s compare our work to two similar papers that conducted a survey study on scientists, both published in highly reputable venues:
>
> |     | **Our Study** |[ Kwon. Nature 2025](https://www.nature.com/articles/d41586-025-01463-8) | [Michael et al. ACL 2023](https://aclanthology.org/2023.acl-long.903) |
> |----------------------------|---------------|----------------------------------------------------------------------------------|--------------------------------------------------------------------------------|
> | **Scope of surveyed disciplines** | Researchers across multiple fields and publication venues | Only Springer Nature authors and newsletter readers | NLP only, primarily ACL conferences |
> | **Response rate** | We believe our true response rate is likely higher because the verified authors email list we used was collected **over 10 years** and many emails may no longer be active. For example, while we sent 107,346 emails, the number of emails opened was 34,922 (or **32.5%**). If we had taken 1226 / 34,922 as the response rate, then it would be 3.5%. The true response rate lies between 1.1% to 3.5%, depending on what proportion of our targeted email addresses are no longer active.We will edit our paper to clarify this. | They used 3 channels to collect survey responses: (i) authors of recent Nature submissions, (ii) Nature Marketing Intelligence researcher panel, and (iii) email recruitment.   The first two had 4.2% and 5.9% response rate, but are narrow recruitment scope. Their email campaign, which is more similar to ours, would have broader scope but they **did not report response rate**. | They recruited through a variety of channels, but mostly Twitter, word-of-mouth, Slack channels, with a small amount of direct email. As such, there is **no way to know response rate**. They performed an analysis estimating number of ACL Anthology published authors and estimated 5% response, but this excludes NLP authors in other venues, so true response rate is likely lower. |
> | **Release survey responses** | Yes | Yes | No|
> | **Release code for statistical analysis** | Yes| No | No
>
> Our work is actually able to provide much more **transparency** around non-response issues, which all surveys must deal with. Unlike other survey studies, we **release our data and statistical analysis code** for future work to analyze in different ways.
>
> While our response rate appears to be low, we target a **broader pool of researchers across fields and venues**. Coverage at increased scale always comes with increasing recruiting effort, which can mean lower reported response rate (larger denominator), but we believe our approach is still highly valuable compared to others’ which target a much smaller pool of researchers (e.g. only one journal/venue, only one discipline) which generally leads to a higher reported response rate (smaller denominator). As such, we believe our observed response was among one of the best that is achievable to study this question across wide fields/demographics.
>
> ## Higher proportion of computer scientists among participants
> As all surveys studies must grapple with this issue, we have mitigated this in several ways:
>
> First, **we controlled potential demographic skew in our statistical models**. We conducted subgroup analyses isolating each demographic (comparisons between CS and other fields, white versus non-white…), to ensure that our findings are not solely driven by the dominant group. Furthermore, several of our findings are specific to or vary across demographic groups, and our discussions are sensitive to and reflective of these differences.  That being said, we agree with the reviewer that more information about the platform itself will help contextualize our results; as such, **we will include more platform demographic statistics**, such as a field of study breakdown from a random sample of 752 verified claimed authors from the research platform (e.g. Computer Sci: 50.1%, Bio & Med: 23%, Social Sci & Humanities: 11.5%, Natural Sci & Engineering: 15.4%).
>
> Second, **we are completely transparent about our process and results**. We share our survey questions, data, and the analysis code. Finally, we acknowledge the unbalanced demographics in our sample (line number 322 - 329).

---

> > ### Comment · Reviewer_5NYA · 2025-06-04
> >
> > Thank you for your response and clarification. I acknowledge the transparency:
> >
> > "Our work is actually able to provide much more transparency around non-response issues, which all surveys must deal with. Unlike other survey studies, we release our data and statistical analysis code for future work to analyze in different ways."
> >
> > But I disagree that this transparency solves the potential biases. However, the addition of:
> >
> > "we will include more platform demographic statistics, such as a field of study breakdown from a random sample of 752 verified claimed authors from the research platform (e.g. Computer Sci: 50.1%, Bio & Med: 23%, Social Sci & Humanities: 11.5%, Natural Sci & Engineering: 15.4%)."
> >
> > helps to contextualize the sample from the paper:
> >
> > "Field of Study: Computer Science (40%); Social Science & Humanities (24%); Natural Science & Engineering (21%); Biology & Medicine (15%)."
> >
> > I believe it is a good addition to the paper. However, the authors should give more details on how they draw this random sample.

---

> > > ### Author Response · Authors · 2025-06-07
> > >
> > > Thank you for acknowledging the transparency of our survey results and analysis process. The random sample is drawn from a SQL database maintained by the research platform.

---

### Official Review · Reviewer_iVRx · 2025-05-09

**Rating:** 7
**Confidence:** 3
**Ethics Flag:** 1

**Summary:**

The paper presents a survey on how LLMs are used as tools for research. There have been previous studies on this topic, but the authors claim that those have either focused on a specific domain or relied on small sample sizes, while the current study relies on researchers from different fields of knowledge and with varied profiles, and uses a larger sample size of 816. The survey participants were sourced from a "large research platform", which remains unspecified.

Usage of LLMs is categorized under six types, namely Information Seeking, Editing, Ideation & Framing, Direct Writing, Data Cleaning & Analysis, and Data Generation. For each type, the participants are asked about their LLM usage frequency and perception, using a five-point Likert scale.

The main findings are that most (81%) respondents use LLM in some way, mainly for search and for text rewriting, and the existence of statistically significant differences between demographics.

**Questions To Authors:**

Why don't you explicitly name the research platform from where you sourced the participants? Is it an anonymization issue? But, if so, why didn't you do the same thing you did when naming the IRB that reviewd the study and explicitly used "omitted for anonymity" (cf. line 107)?

As mentioned above, the participant demographics are skewed, but this may simply be a reflection of a skew that is (almost certainly) already present in the members of the research platform that you used as a source. Do you any any demographic statistics on the membership of this platform?

Some other feedback below (as there's no specific section for this):
- 115: Having "Non-Native English" under "Native Language" sounds very strange. I'd expect either (a) Native Language: English / Other, or (b) Language: Native English / Non-native English.
- 651: "Prefer not to disclose" appears twice
- Figure 6: "Prefer to self-describe:"

**Reasons To Accept:**

- While there are some issues with the participant's data (see below), it is good to have some quantitative data on this issue.
- The data is made available.

**Reasons To Reject:**

- The participants are self-selected: The recruitment email was sent to over 100k people, with only 1226 responses having been collected. The ones that choose to participate, whatever their motivation, may not be representative of the whole.
- The data is self-reported: Participants can very well admit to using LLMs for small adjustments to writing, but never to, say, fully write a paper or a review, but we can't really know whether they are being honest. Who would admit to unethical behaviour on a survey? Even if you, the study authors, claim it is anonymous, the participant may nonetheless be afraid to risk disclosing this information.
- The demographics may not be representative: The participants come from different fields of study, but the distribution is nevertheless quite skewed towards STEM domains, with only 24% from Social Sciences and Humanities. The demographic profiles of the participants are also skewed towards white (61%), male (79%), etc.
- The data may show an outdated picture: The data was collected over a six month period from November 2023 to April 2024, so it's at least one year old. The capabilities of LLMs are progressing so quickly, leading to big shifts in perception on what LLMs can do, that this may be a bit too old to have an accurate account of how LLMs are currently perceived.

---

> ### Author Response · Authors · 2025-06-03
> **Response to survey participants are self-selected & data is self-reported.**
>
> Thank you for your review and feedback. Our responses to your review are as follows:
> ## Survey participants are self-selected & data is self-reported.
> Response bias and self-reporting bias are inevitable for any survey study. We believe we have approached this in a manner that places our survey methodology among the most rigorous within this space. For example, let’s compare our work to two similar papers that conducted a survey study on scientists, both published in highly reputable venues:
> |     | **Our Study** |[ Kwon. Nature 2025](https://www.nature.com/articles/d41586-025-01463-8) | [Michael et al. ACL 2023](https://aclanthology.org/2023.acl-long.903) |
> |----------------------------|---------------|----------------------------------------------------------------------------------|--------------------------------------------------------------------------------|
> | **Scope of surveyed disciplines** | Researchers across multiple fields and publication venues | Only Springer Nature authors and newsletter readers | NLP only, primarily ACL conferences |
> | **Response rate** | We believe our true response rate is likely higher because the verified authors email list we used was collected **over 10 years** and many emails may no longer be active. For example, while we sent 107,346 emails, the number of emails opened was 34,922 (or **32.5%**). If we had taken 1226 / 34,922 as the response rate, then it would be 3.5%. The true response rate lies between 1.1% to 3.5%, depending on what proportion of our targeted email addresses are no longer active.We will edit our paper to clarify this. | They used 3 channels to collect survey responses: (i) authors of recent Nature submissions, (ii) Nature Marketing Intelligence researcher panel, and (iii) email recruitment.   The first two had 4.2% and 5.9% response rate, but are narrow recruitment scope. Their email campaign, which is more similar to ours, would have broader scope but they **did not report response rate**. | They recruited through a variety of channels, but mostly Twitter, word-of-mouth, Slack channels, with a small amount of direct email. As such, there is **no way to know response rate**. They performed an analysis estimating number of ACL Anthology published authors and estimated 5% response, but this excludes NLP authors in other venues, so true response rate is likely lower. |
> | **Self-reporting bias** | 100% responses from verified authors                                                                                                                                                                                                                                                                                                                                                                                                         | 51.2% responses from verified authors. Rest were recruited from a public newsletter and **relied on self-reporting**                                                                                                                                  | Given the emphasis on social media profiles, almost all results are based on **self-reporting** without verification. Only in the 568 emails sent to senior authors were identities manually verified.|
> | **Release survey responses** | Yes                                                                                                                                                                                                                                                                                         | Yes                                                                                                                                                                                                                                                                 | No                                                                                                                                                                                                                                                       |
> | **Release code for statistical analysis** | Yes                                                                                                                                                                                                                                                                                                                                                                                                                                                        | No                                                                                                                                                                                                                                                                  | No
>
> See next comment for more response to this review.

---

> > ### Author Response · Authors · 2025-06-03
> >
> > ## (Continued) Survey participants are self-selected & data is self-reported.
> >
> > Our work is actually able to provide much more **transparency** around non-response issues, which all surveys must deal with. Unlike other survey studies, we **release our data and statistical analysis code** for future work to analyze in different ways.
> >
> > All surveys must deal with self-reporting bias.  We mitigate this in different ways:  First, we use **100% verified authors** which validates the identities of the authors and decreases chances of lying compared to anonymous online forms, especially for demographic data. Second, we target a **broader pool of researchers across fields and venues**. Coverage at increased scale always comes with increasing recruiting effort, which can mean lower reported response rate (larger denominator), but we believe our approach is still highly valuable compared to others’ which target a much smaller pool of researchers (e.g. only one journal/venue, only one discipline) which generally leads to a higher reported response rate (smaller denominator). As such, we believe our observed response was among one of the best that is achievable to study this question across wide fields/demographics.
> >
> > In all, while we appreciate the reviewer’s concern with data bias, many of these are inherent disadvantages of surveys overall. Despite this, we believe surveys are the appropriate methodology for this work:
> > - First, **survey is a useful methodology compared to alternatives**: For example, directly observing researchers’ LLM usage at scale is infeasible outside of accessing user data at a company product, and even then, such results may not have a path to public sharing. Another example is in-person, non-anonymized, small-scaled interviews, which are also not without their own biases (e.g. lying, unnatural behavior due to study conditions).
> > - Second, while it is true that people may lie when answering surveys, **the possibility of less candid responses does not undermine the necessity of self-reported data collection**, particularly when the research goal is to understand the perceptions of the target community. Whether a participant understates their LLM use due to perceived stigma or overstates it to appear technologically savvy, we believe that both responses are meaningful data that illuminate the ethical and cultural tensions surrounding AI use in academic work.
> > - Lastly, recognizing the potential stigma surrounding LLM use in certain fields (particularly in the humanities, as indicated by survey responses), **we sought to establish an environment where participants could respond openly and honestly**. To ensure anonymity, we collected only verified email addresses through the research platform and deliberately avoided linking responses to any other author information available on the platform. For these reasons, we did not collect any Personally identifiable information (PII) and participants were also free to use incognito mode to complete the survey. We will address this in the limitations.
> >
> > ## Participant demographics may be skewed towards certain groups
> > As all surveys studies must grapple with this issue, we have mitigated this in several ways:
> >
> > First, **we controlled potential demographic skew in our statistical models**. We conducted subgroup analyses isolating each demographic (comparisons between CS and other fields, white versus non-white…), to ensure that our findings are not solely driven by the dominant group. Furthermore, several of our findings are specific to or vary across demographic groups, and our discussions are sensitive to and reflective of these differences.  That being said, we agree with the reviewer that more information about the platform itself will help contextualize our results; as such, **we will include more platform demographic statistics**, such as a field of study breakdown from a random sample of 752 verified claimed authors from the research platform (e.g. Computer Sci: 50.1%, Bio & Med: 23%, Social Sci & Humanities: 11.5%, Natural Sci & Engineering: 15.4%).
> >
> > Second, **we are completely transparent about our process and results**. We share our survey questions, data, and the analysis code. Finally, we acknowledge the unbalanced demographics in our sample (line number 322 - 329).
> > The reviewer has asked why we didn’t name the research platform. Indeed, **we omitted this research platform due to anonymity**. We will share the platform after the double blind review process.

---

> > > ### Author Response · Authors · 2025-06-03
> > >
> > > ## The survey data may be outdated
> > > We acknowledge the survey is one year old, and that sentiment and adoption rate in the community are likely changing quickly. However, we think one year is within a reasonable timeline to collect, clean, analyze data, and write up the results for publishing. More importantly, even if exact effect sizes on sentiment and adoption rate change over time, we believe our core research findings—especially those about differences across demographics—are likely to hold stable trends over time. Finally, **we will completely open our surveys, results, and statistical analysis scripts** so that the community can reexamine the state of our community in the future, and the current results provide a **valuable baseline data point in time for future comparison**, something that other survey studies have not done.

---

> > > > ### Comment · Reviewer_iVRx · 2025-06-06
> > > > **Reviewer comment to responses above**
> > > >
> > > > I'll add a comment only here, but also address the responses above.
> > > > I understand that some of the issues I pointed out may be inevitable in such a survey, but they remain weaknesses. Nevertheless, putting your survey in context by comparing it with similar studies is quite informative, as well as the clarification of the response rate and the platform statistics. Including some of this information in the final paper will help to better put it into context.
> > > > Overall, the responses have slightly eased some of my concerns. Accordingly, I'll raise the score from 6 to 7.

---

### Decision · Program_Chairs · 2025-07-08

**Decision:**

Accept

**Comment:**

This is the result and analysis of a large-scale survey (816 participants) for researchers, inquiring over the use of LLMs in their research process. Some of the more interesting findings are the high prevalence as well as the difference perception of their utility and concerns across demographic traits

The discussion was mostly centered on the drawbacks of relying on self-reported metrics, as well as the impact this self-selection had on the demographic distribution and the answers per se. Many of these are intrinsic to the nature of this work, and the authors have done a good job in replying to those concerns

This is certainly an interesting paper which will fuel good discussions at COLM and beyond. It should be reminded that this is a snapshot - redoing this survey now (a year later) might yield very different results. At the same time, it is this snapshot which might prove impactful in the future when studying the trajectory of LLM usage in research. So many things are changing at the same time, that having some point analysis might become very influential

[Automatically added comment] At least one review was discounted during the decision process due to quality]